# Using RothC Model to Simulate Soil Organic Carbon Stocks under Different Climate Change Scenarios for the Rangelands of the Arid Regions of Southern Iran

**Sayed Fakhreddin Afzali [1,\*], Bijan Azad [2], Mohammad H. Golabi [3] and Rosa Francaviglia [4]** 

1 Department of Natural Resources and Environmental Engineering, College of Agriculture, Shiraz University, Shiraz 71441-65186, Iran

2 Department of Rehabilitation of Arid and Mountainous Regions, Faculty of Natural Resources, University of Tehran, Tehran 31587-77871, Iran; bijanazad@ut.ac.ir

3 Soil Sciences, College of Natural and Applied Sciences, University of Guam, Mangilao, Guam-USA, GU 96923, USA; mgolabi@triton.uog.edu

4 CREA, Council for Agricultural Research and Economics, Research Centre for Agriculture and Environment, 00184 Rome, Italy; rosa.francaviglia@crea.gov.it

\* Correspondence: afzalif@shirazu.ac.ir

**Abstract:** Soil organic carbon (SOC) is strongly influenced by climate change, and it is believed that increased temperatures might enhance the release of $CO_2$ with higher emission into the atmosphere. Appropriate models may be used to predict the changes of SOC stock under projected future scenarios of climate change. In this investigation, the RothC model was run for a period of 36 years under climate scenarios namely: P (no climate change) as well as CCH1 and CCH2 (climate change scenarios) in the arid rangelands of Ghir–O-Karzin's BandBast in southern Iran. Model results have shown that after 11 years (2014–25), SOC stock decreased by 3.05% under the CCH1 scenario (with a projected annual precipitation decrease by 6.69% and mean annual temperature increase by 9.96%) and by 0.23% under the P scenario. In CCH2, with further decreases in rainfall (10.93%) and increase in temperature (12.53%) compared to CCH1, the model predicted that the SOC stock during the 25 years (2025–50) was reduced by 2.36% and 3.53% under the CCH1 and CCH2 scenario respectively. According to model predictions, with future climatic conditions (higher temperatures and lower rainfall) the decomposition rate may increase resulting in higher losses of soil organic carbon from the soil matrix. The result from this investigation may also be used for developing management techniques to be practiced in the other arid rangelands of Iran with similar conditions.

**Keywords:** global warming; soil carbon model; rangelands of southern Iran

## 1. Introduction

Carbon stored in soils is the largest carbon pool in terrestrial ecosystems, this is twice the amount of carbon in the atmosphere and three times the amount of carbon in the biotic world [1]. Therefore, soils have a great role in maintaining the balance of the global carbon cycle [1]. In general, carbon sequestration depends on carbon equilibrium and is affected by abiotic factors and management practices [2]. As reported by the authors of Reference [3], any small changes in the soil organic carbon (SOC) will have a major impact on the concentration of $CO_2$ in the atmosphere, hence affecting the climate parameters.

Based on the mean annual rainfall, regions receiving 200–500 mm of winter rainfall are defined arid [1]. The soils in the arid regions have a low carbon content and considering the extent of these areas [4] they may have a special place in carbon sequestration [2]. The rangelands in the arid regions,

which cover about 40% of Earth's surface area can sequester high amount of $CO_2$ in the soil due to their prevalence around the globe [5]. However, in these ecosystems the poor vegetation cover, the low concentration of litter, the sparse vegetation, and the low biodiversity of plants species, lead to low SOC content [2,6].

Climate factors such as precipitation and temperature have a significant effect on SOC contents which is the basic component of the global carbon cycle, particularly in the context of climate change [7–11]. Most researchers [10,12–14] have reported losses from the soil organic carbon under the climate change conditions. In Reference [15], the authors stated that increasing temperature lead to positive feedback between climate change and carbon cycle and more losses SOC to the atmosphere in form of $CO_2$ and accelerated global warming. Some other studies [16] have reported reduction in soil organic decomposition under conditions of future climate change and global warming scenarios. However, the impacts of projected future scenarios of climate change on SOC and its dynamics are still largely uncertain [17]. It is therefore crucial evaluating the impact on SOC stocks and their dynamics under projected future scenarios of climate change.

Because of complexity of the soil-plant-atmosphere system, simulation models can be useful for studying this relationship within the aforementioned systems and also predicting the changes of SOC stock under projected future scenarios of climate change [12]. Many models, such as RothC, Daisy, DNDC and Century have been developed for this purpose. Specifically, RothC is one of the models [18] widely used to assess the effects of future climate change on the SOC dynamics [9,12,13,19].

Researchers in Reference [20], using RothC reported that soil organic carbon stocks decreased, due to increased decomposition rates as a result of higher temperatures under climate change conditions in Ukraine and Russia. The authors of Reference [13], using the RothC model, assessed the effects of climate scenarios on soil carbon changes in a rangeland ecosystem of southern Ireland. Wan et al. [12] using the RothC model have predicted that SOC stock will decrease under climate change scenarios in China. Reference [21] investigated the effects of climate change on soil carbon changes in a Mediterranean ecosystem using the RothC model. Authors of Reference [22] stated that the trend of SOC loss in form of $CO_2$ under climate change is increasing in dryland ecosystems of Australia.

About 70% of Iran's rangelands are located in the arid and semi-arid regions of the country. Rangelands of Iran cover 85 million ha of Iran's lands and about 51% of country's area. Ghir–O-Karzin's BandBast is one of the most important arid regions of southwest Iran, where rising temperature and shifting precipitation patterns are predicted to yield the highest decrease in rainfall during hot summers and even cold winters [23]. On the other hand, the climate factors are expected to have a major impact on SOC stocks in the aforementioned areas [9,10]. The effect of climate change on SOC dynamics in arid rangelands of Iran has not been adequately quantified, though rangelands occupy a vast land area of the country and might play a main role in climate change mitigation through carbon sequestration in soils. On the other hand, previous studies showed positive [16] or negative effects of climate change [12,20] on SOC stocks, but the results of these studies cannot be generalized to other regions, especially arid rangelands. Besides forage production, the arid rangelands of Iran provide many important ecosystem services including climate and water regulation, and opportunities for SOC sequestration because of their low carbon content. So far, no studies have examined SOC stock dynamics with simulation models in arid rangelands of Iran, and the present study is the first attempt to understand the effects of climate change on SOC stock. The main purposes of this study were to: (1) validate the RothC model simulations with the measured SOC stocks and (2) to determine the effects of different scenarios of climate change on SOC dynamics in the arid rangelands of Ghir–O-Karzin's BandBast in southwest of Iran. In this regard our research is expected to provide the first predictions of SOC stock under different climate change scenarios in the south of Iran.

## 2. Materials and Methods

### 2.1. Study Area

The BandBast rangeland in the south Iran includes about 2380 hectares and is located about 200 km southeast of Shiraz (29′36″ N, 52′32″ E) in southern Iran (Figure 1). Based on the 32-year (1983 to 2014) average data obtained from the meteorological station in Ghir–O-Karzin, total annual precipitation is 275.36 mm, mean annual temperature is 23.94 °C, and total annual open-pan evaporation rate is 2910.98 mm (Figure 2). Generally, the local climate is warm to very hot during summers and relatively humid during the winters. The BandBast rangelands have Alluvio-Colluvial soil properties and lie within a relatively flat basin physiography where mean elevation is 700 meters above sea level (m.a.s.l). Soils are mainly Entisols based on soil taxonomy [24] with silty textures. The average of soil organic carbon (%), electrical conductivity (ds m$^{-1}$), and total nitrogen (%) in the top soil (20 cm) are 0.62, 0.66 and 0.06, respectively. The natural vegetation includes annual and perennial grasses of C$_3$ type as potential native vegetation, with low percent of cover and low biomass production. *Agropyron* sp., *Dactylis* sp., *Bromus* sp., *Hordeum* sp., *Carex* sp. and *Seidlitzia* sp. have been observed predominantly within this region. Historically the BandBast rangelands are managed with light grazing in the spring and no management practices such as fertilization are adopted.

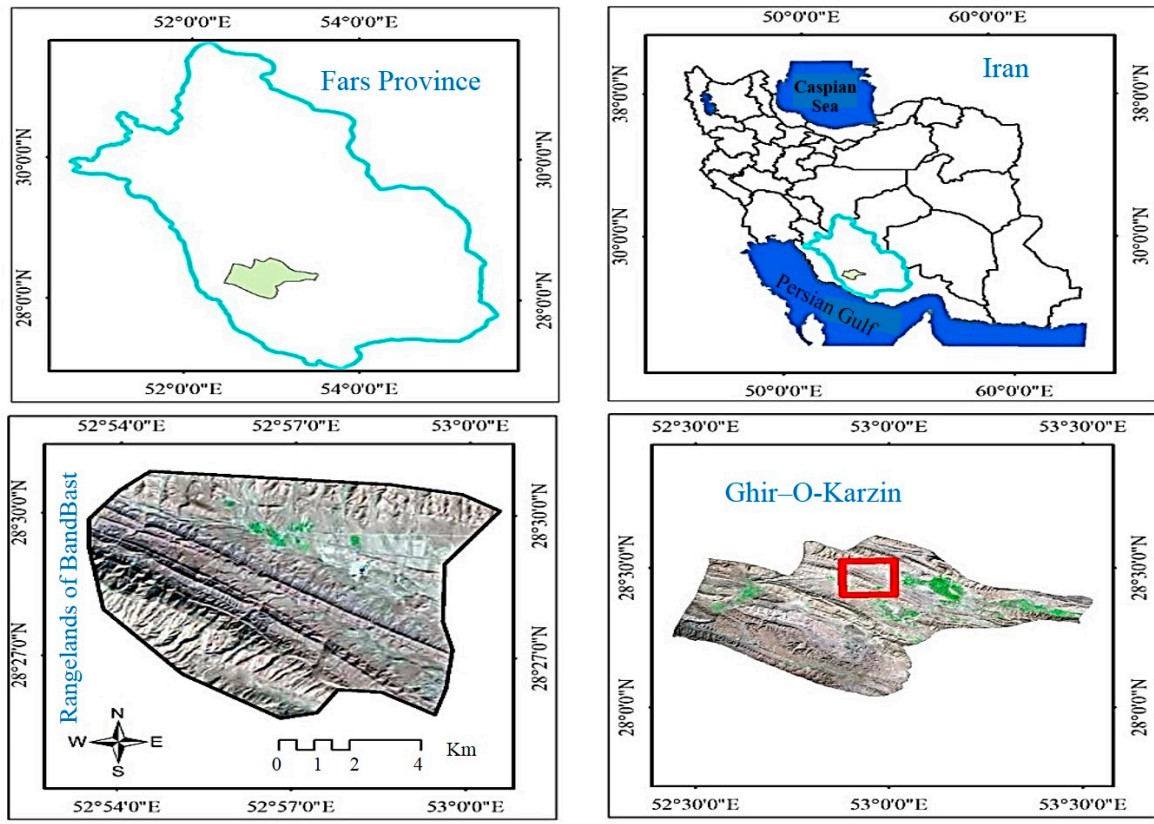

**Figure 1.** Location map of the BandBast rangelands in the south Iran.

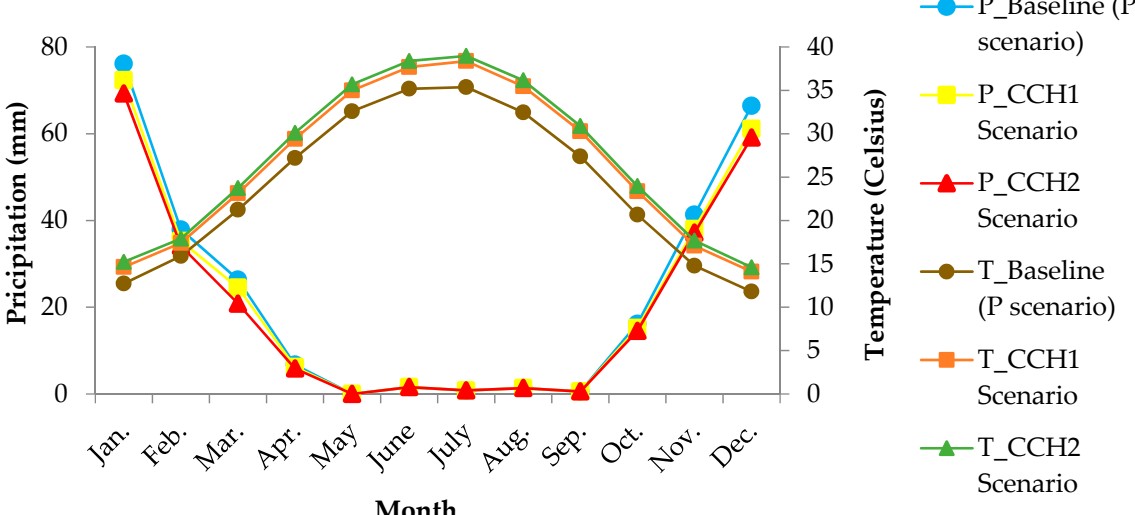

**Figure 2.** Change in monthly precipitation (P), and monthly air temperature (T) under three climate scenarios, P, CCH1 and CCH2.

### 2.2. Sample Collection and Field and Laboratorial Measurements

Collection of soil samples from BandBast rangelands was carried out randomly in order to avoid bias, during four successive months (April to July of 2014) of sampling. Due to the homogeneity of the region in terms of topography and geology, soil samples were collected based on a simple random sampling design and sampling points were irregularly distributed. Soil sampling was done for four successive months and after removing the litter layer, 20 samples were taken each month (for a total of 80 samples) within the 0–20 cm depth of the soil. To determine the soil bulk density, two soil core samples (in addition to routine soil samples) were collected in each sampling point. The soil samples were air dried, visible plant material were removed from the samples, and sieved through a 2-mm sieve for further analysis. The soil texture and soil organic carbon (%), were measured by Hydrometry method [25] and Walkley and Black method [26] respectively. Using the Core method [27] the soil bulk density was measured in the laboratory. Finally, the SOC stock was calculated for the 0–20cm soil layers using the following equation [28]:

$$\text{SOC Stock} = \text{SOC content } (\%) \times \text{Layer thickness (cm)} \times \text{bulk density} \left( \text{g cm}^{-3} \right) \tag{1}$$

### 2.3. RothC Model

The RothC model has four active SOC pools [18,29] namely; decomposable plant material (DPM), resistant plant material (RPM), microbial biomass (BIO), and the humified organic matter (HUM). The RothC model also includes an inert organic matter (IOM) pool which is resistant to decay. The IOM however has, a nominal radiocarbon age of 50,000 years which does not take part in C turnover. The RothC model partitions incoming plant residues into resistant plant material (RPM) and decomposable plant material (DPM), depending on the DPM:RPM ratio of the particular plant material. The plant material decomposes to form $CO_2$, biomass (BIO) and humified organic matter (HUM). All active pools undergo decomposition by first-order kinetics, each with a characteristic rate of decomposition. This rate is modified according to soil moisture, temperature and the soil surface vegetation cover according to the conditions of a particular month of the year. Soil clay content also affects the partition between $CO_2$ evolved and BIO + HUM formed. In the present study, the RothC model version 26.3 [18] was used to simulate SOC dynamics. The input data to run RothC model are included in two files (weather and land management), each containing a set of variables. The model is designed to run in two modes: "*forward*" using known carbon inputs to calculate the changes in soil

organic carbon and "*inverse*", when carbon inputs are calculated at equilibrium state from known soil organic carbon contents.

### 2.4. Inputs Data and Model Calibration

Inputs data to run RothC model include data on climate (average monthly precipitation, monthly air temperature and monthly open-pan evaporation), soils (soil organic carbon content, clay content, soil depth, soil bulk density) and vegetation data (land use type, DPM:RPM ratio, the amount of carbon returned into the soil from the plant litter and soil surface cover during the year). Climate data were obtained from the 32-year of the meteorological at the Ghir–O-Karzin station in southern Iran. Soil data were determined using field and laboratory measurements (Table 1). DPM/RPM ratio or litter quality factor in rangeland ecosystems of the Ghir–O-Karzin (0.67) was set to the default values for grasslands [18] (Table 1). The RothC model was run at equilibrium (*inverse mode*) to calculate the amount of carbon returned to the soil from the annual plant residue, based on the known total soil carbon content, clay content, the inert organic matter (IOM), climatic conditions as well as the soil surface cover (Table 2). The model derives IOM for the 0–20 cm of the soil layer (Table 1) by using the following equation [30]:

$$IOM = 0.049 \, (\text{total SOC})^{(1.139)}$$

Model calibration was done based on a procedure adopted in previous studies [11,21], i.e., calibrating RothC at equilibrium state under the potential native vegetation by running the model in *inverse mode*. The model was run with climate, vegetation and soil conditions in rangelands of Ghir–O-Karzin's BandBast at equilibrium state for 10,000 years ending in January 2002. The equilibrium condition represents the baseline for the evaluation of the management and climate change effects on soil organic carbon in the ecosystem. The calibration process run the model and automatically adjusted plant carbon inputs until simulated SOC stock were equal to the measured SOC stock (Table 2) in January 2002 [11,18,19,21,31]. Table 1 lists some of the parameters used in the model based on measured SOC stock in January 2002 that was derived from the study of Reference [32].

**Table 1.** Soil and vegetation characteristics and climatic parameters in BandBast rangelands.

| Parameter | Value |
|---|---|
| Location | 52°59′ N, 29°28′ E |
| Total precipitation (mm) | 275.36 |
| Mean temperature (°C) | 23.94 |
| Total open-pan evaporation (mm) | 2910.98 |
| Texture (sand, silt, clay) | 36%; 51%; 13% |
| Bulk density (gr/cm$^3$) | 1.34 |
| pH | 8 |
| Initial total soil organic carbon (Mg C ha$^{-1}$) | 16.68 |
| Soil depth (cm) | 20 |
| Soil type | Entisols |
| Farmyard manure C inputs (FYM) (Mg C ha$^{-1}$) | 0 |
| Historical land use/native vegetation | Rangeland |

**Table 2.** Input data, measured and modeled SOC stock in 2002 (mean values of the samplings in January 2002) in the native vegetation.

| Vegetation Cover | Clay (%) | Inert Organic Matter (IOM) (Mg ha$^{-1}$) | Modeled Soil C Inputs (Mg C ha$^{-1}$) | DPM/RPM [a] | Measured SOC Stock (Mg C ha$^{-1}$) | Modeled SOC Stock (Mg C ha$^{-1}$) | Deviation (%) [b] |
|---|---|---|---|---|---|---|---|
| Rangeland | 13 | 1.2086 | 0.9561 | 0.67 | 16.68 | 16.68 [c] | 0.00 |

[a] Decomposable Plant Material/Resistant Plant Material; [b] Deviation (%) calculated as [100 × (modeled-measured)/measured]; [c] Model run to the equilibrium in "inverse mode".

### 2.5. Validation of RothC Model

The RothC model was validated in the "*forward mode*" using the SOC stock data obtained from this study. Specifically, we compared the model output to a set of data independent from the calibration stage, i.e., the 4 monthly data from April to June 2014. Some statistical comparisons between the simulated and measured data based on determination factor ($R^2$), root mean square error (RMSE) and performance efficiency (PE) were used for model validation (Equations (2) and (3)). The smallest value for RMSE is zero, indicating that there is no difference between measured and simulated values. The PE (Modeling Efficiency) has values ranging from $-\infty$ to 1. The model's best performance is at PE = 1. RMSE and PE were defined as:

$$\text{RMSE} = \sqrt{\frac{\sum_{i=1}^{n}(O_i - P_i)^2}{n}} \tag{2}$$

$$\text{PE} = 1 - \frac{\sum_{i=1}^{n}(P_i - O_i)^2}{\sum_{i=1}^{n}(O - \bar{O})^2} \tag{3}$$

where $O_i$ and $P_i$ are observed data and the predicted SOC, $\bar{O}$ are the mean values from the observed data, and n is the number of the paired values.

### 2.6. Scenarios of Climate Change

According to the forecasts, Ghir–O-Karzin is expected to experience an increase in temperature and a decrease in rainfall amounts under predicted climate change conditions [23]. Using a General Circulation Model (UKMO), Koocheki et al. [23] have projected a mean yearly increase in temperature by 9.96% and 12.53% and an annual reduction in rainfall by 6.69% and 10.93% for the years 2025 and 2050 respectively. Climate scenarios used in this study were: P scenario ('no climate change' conditions or present climate condition based on the average monthly rainfall and mean monthly temperature during the period 1983–2014), CCH1 scenario (climate change conditions with a projected annual rainfall decrease by 6.69% and a mean annual temperature increase by 9.96%) and CCH2 scenario (climate change conditions with a further decrease by 10.93% in rainfall and increase by 12.53% in temperature compared to the scenario CCH1) (Table 3). The P scenario is classified into three sub-scenarios of $P_{14-25}$, $P_{14-50}$ and $P_{25-50}$ whose respective climate conditions refer to the periods 2014–25, 2014–50 and 2025–50. The CCH1 scenario on the other hand refers to the climate change conditions for 2025, and the CCH1$_{25-50}$ scenarios was considered to provide climate condition for years of 2025–50. The CCH2 scenarios provide condition of climate change from 2025 to 2050, but with further decreases in rainfall and increase in temperature compared to the scenario CCH1. The values listed in Table 3 are used as input to the RothC model for the 2025 and 2050 period. It should be noted that due to the structure of the RothC model, the meteorological data (including climate change scenarios for future) is static for each period (for example at 2025 to 2050). That means that the value of each meteorological parameter (rainfall, temperature, and evaporation) for a given month will enter into the RothC model in the long-term simulation. The total monthly open-pan evaporation was calculated using Penman method for the year 2025 and 2050. Finally, the SOC stock change was simulated under one baseline scenario (P) and two climate change scenarios (CCH1 and CCH2). Statistical analysis was done with the least significant difference (LSD) test.

**Table 3.** Average monthly temperature (°C) and average monthly rainfall (%) based on UKMO for 2025 and 2050 compared to the long-term average in Ghir–O-Karzin.

|  | Year | Jan | Feb | Mar | Apr | May | Jun | Jul | Aug | Sep | Oct | Nov | Dec |
|---|---|---|---|---|---|---|---|---|---|---|---|---|---|
| Temperature (°C) | 2025 | 1.9 | 1.5 | 1.9 | 2.2 | 2.4 | 2.5 | 3 | 3 | 2.9 | 2.7 | 2.3 | 2.3 |
|  | 2050 | 2.5 | 2 | 2.5 | 2.9 | 3.1 | 3.2 | 3.6 | 3.7 | 3.5 | 3.3 | 2.9 | 2.8 |
| Rainfall (%) | 2025 | −5 | −7 | −7 | −8 | −9 | - | - | - | - | −6 | −8 | −8 |
|  | 2050 | −9 | −10 | −21 | −13 | −41 | - | - | - | - | −10 | −10 | −11 |

## 3. Results

### 3.1. Calibration of the RothC Model

The RothC model was calibrated at equilibrium state based on a procedure adopted in previous studies [13,24] and 40 soil samples collected in January 2002 were available from a previous study [32]. These 40 samples represented the original levels of SOC stock in these rangelands to be used in the RothC model. In this research when the RothC model was calibrated at equilibrium state, the simulated value was equal to the measured value of 16.68 Mg C ha$^{-1}$ in January 2002. Based on *inverse mode* simulation, a carbon input to the soil of 0.9561 Mg C ha$^{-1}$ (Table 2) was required to achieve the SOC stock of 16.68 Mg C ha$^{-1}$ to a depth of 20 cm measured in January 2002. Having set the equilibrium conditions in this way for the native vegetation (arid rangeland), the model was run for future periods in the *forward mode* (1983–2014) and the outputs of these simulations will be used for model validation.

### 3.2. Validation of the RothC Model

The SOC stock data obtained from this study (April to July 2014) were used for model validation. A significant linear relationship ($R^2$ = 0.76) was found between the measured total SOC stock and the simulated values (Figure 3). The root mean square error, indicating the total difference between the measured and simulated values was RMSE = 0.014, and performance efficiency was PE = 0.69 (Figure 3). This confirms that the RothC model accurately simulates the dynamics of SOC stock in the BandBast rangelands in southern Iran. The RothC performed well in predicting SOC of the rangelands of Ghir–O-Karzin's BandBast (Figure 3), and the measured and simulated SOC stock values were closely distributed near the 1:1 line.

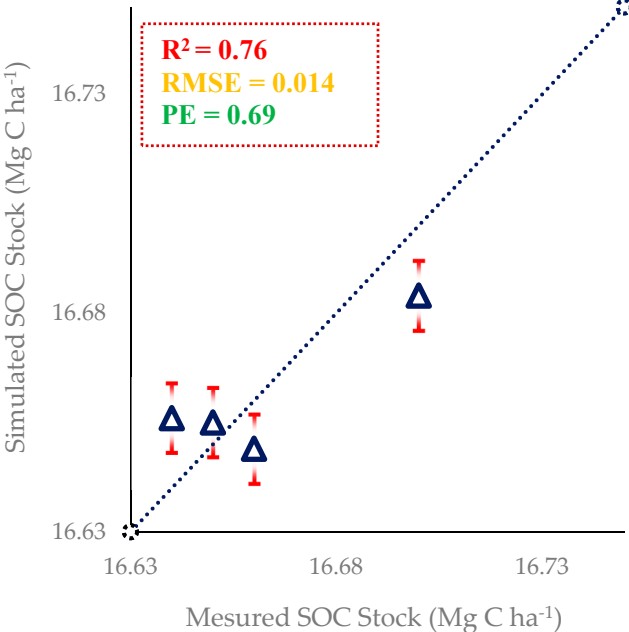

**Figure 3.** Measured and simulated SOC Stock in BandBast rangelands compared with 1:1 line.

### 3.3. Simulation of the SOC Stock under Different Climate Change Scenarios

In the BandBast rangelands, the projected annual precipitation and the mean yearly temperature for 2014–25 (CCH1 scenario) and 2025–50 (CCH2 scenario) periods decreased by 6.69 and 10.93% and increased by 9.96 and 12.53% compared with the long-term values (mean annual precipitation 275.36 mm and mean monthly temperature of 23.94 °C) respectively (Figure 2). The actual SOC stock in December 2014 was 16.68 Mg C ha$^{-1}$ and the value simulated by the model showed that after 11 years (2014–25) the SOC stock had decreased by 3.05% (16.17 Mg C ha$^{-1}$) under the CCH1 scenario and

by 0.23% (16.72 Mg C ha$^{-1}$) under the P scenario (Figure 4 and Table 4). Additionally, there was a significant difference between the P and CCH1 scenarios during 2014 until 2025 ($p < 0.01$) (Figure 5).

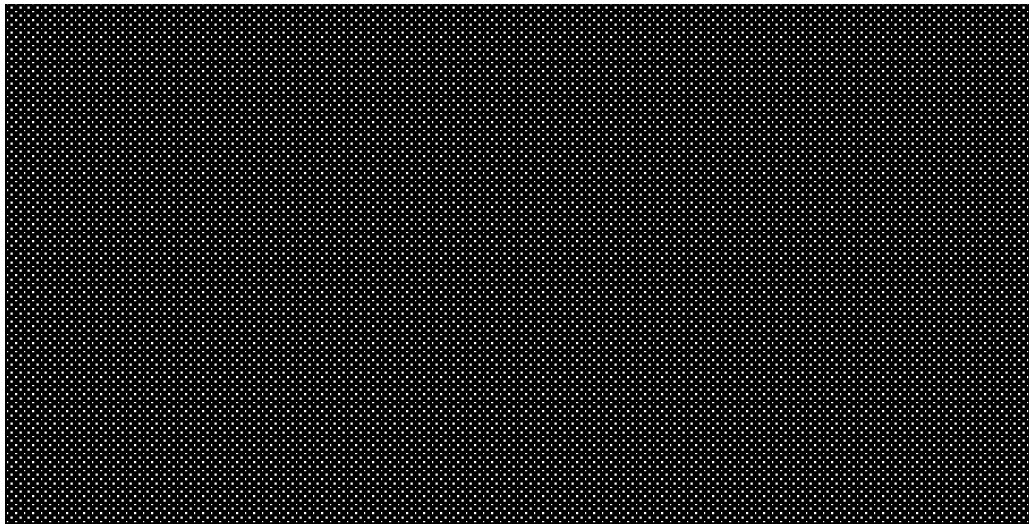

**Figure 4.** Simulated yearly SOC dynamics under three climate scenarios (P, CCH1 and CCH2).

**Table 4.** Losses of SOC stock under three climate change scenarios (P, CCH1 and CCH2) from 2025 to 2050 in the arid rangelands of BandBast.

| Scenario | P | | | CCH1 | | | CCH2 |
|---|---|---|---|---|---|---|---|
| Period | 2014–25 | 2025–50 | 2014–50 | 2014–25 | 2025–50 | 2014–50 | 2025–50 |
| SOC stock (Mg C ha$^{-1}$) | 16.72 | 16.72 | 16.72 | 16.352 | 16.95 | 16.07 | 15.82 |
| SOC loss (Mg C ha$^{-1}$) | 0.04 | 0.01 | 0.04 | 0.51 | 0.38 | 0.89 | 0.57 |
| % of initial SOC | −0.23 [a] | −0.05 | −0.24 | −3.05 | −2.36 | −5.36 | −3.53 |

[a] percentage of SOC change according to the initial (i.e., 2014) SOC stock.

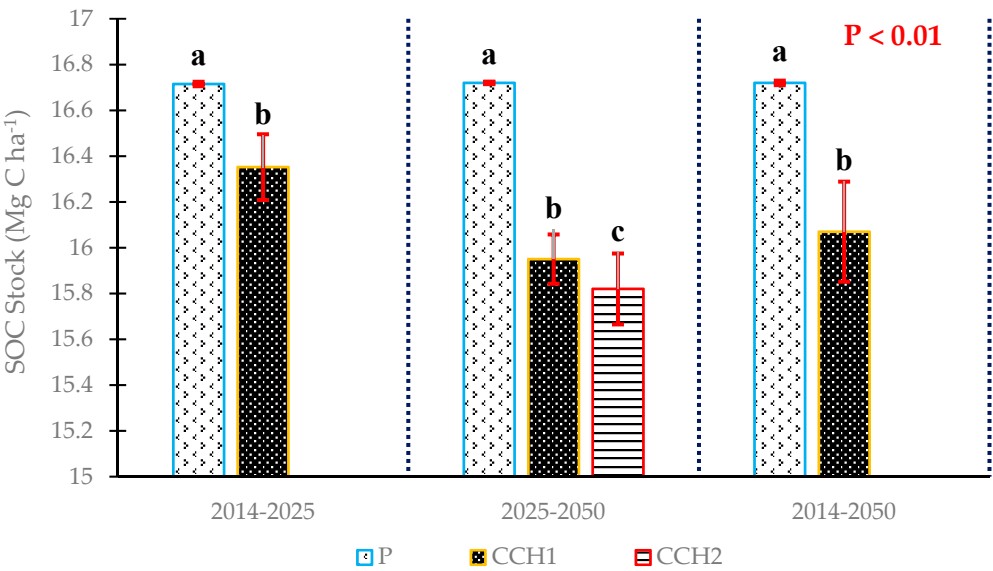

**Figure 5.** SOC stock for three periods 2014–25, 2025–50 and 2014–50 under different climate scenarios. Different superscript letters represent statistically significant differences between climate scenarios at $p < 0.01$ according to the least significant difference (LSD) tests.

With the highest increase in temperature and the highest decrease of rainfall, from CCH1 to CCH2 (for 2025–50 period) (Figure 2), SOC stock decreased from 16.17 to 15.60 Mg C ha$^{-1}$ under CCH2

scenario during 2025 until 2050 (Figure 4). Long-term changes of SOC stocks in response to climate change from 2025 until 2050 showed significant differences ($p < 0.01$) (Figure 5), and the minimum (0.05%) and maximum (3.53%) losses of SOC stock were observed in the P and CCH2 scenarios (Table 4). Additionally, with continued climatic condition of CCH1 scenario from 2025 until 2050, SOC stock decreased by 2.36% (Table 4). The trend of soil organic carbon stock during the 25 years (2025–50) was decreasing under the CCH1 and CCH2 scenario (Figure 4). Generally, at higher temperature and also lower precipitation, SOC stocks decreased significantly in the two scenarios of CCH1 and CCH2 in comparison with the no climate change (P scenario) scenarios (Figures 4 and 5).

With continued climatic conditions of CCH1 from 2014 until 2050 the SOC stock decreased by 0.24% and 5.36% under the P and CCH1 scenario, respectively (Table 4) and a significant difference ($p < 0.01$) was observed between them (Figure 5). Conversely to the considerable reduction of SOC stocks under the CCH1 and CCH2 scenarios, the trend of SOC stocks under P scenario during 36 years (2014–50) was in a steady-state indicating these rangelands are in equilibrium state (Figure 4).

## 4. Discussion

Measured SOC stocks were well correlated with predicted values obtained from the RothC simulation using the current scenario (P scenarios), with PE values of 0.69 (Figure 3). The results of the model validation indicated that the RothC model has been able to simulate the changes of soil organic carbon (Figure 3). Therefore, the RothC model can be applied for simulating the dynamics of SOC stock in the arid rangelands of BandBast region. Several previous studies [11,21,31] on the performance of the RothC model for the simulation of the soil organic carbon stock have also been consistent with the result of this study.

Our study also showed that the SOC stock during the 36 years of evaluation decreased by 0.23% under the P scenario (Table 4) however the change in comparison with the baseline year (2014) was not significant (Figure 5). We can therefore state that in general terms the BandBast rangelands are in equilibrium conditions [31]. Modeling soil organic carbon stock in the same rangelands, authors of Reference [33] stated that the SOC of rangelands was in equilibrium during the simulation period. In natural conditions, the input of carbon in the form of litter is equal to the output of carbon in the form of $CO_2$ and rangelands are in a state of equilibrium as reported by Reference [34]. Several modeling studies [34] also have shown that the soil carbon content ranges are very close to the amount of carbon that had been measured and recorded in the previous years, when there were no specific management practices implemented.

The climatic scenario CCH1 had a significant impact on SOC stock dynamic when compared to no climate change scenario (P scenario) (Figures 4 and 5). There was an overall decreasing trend of SOC stocks in the CCH1 scenario (Figures 4 and 5), and in the arid rangelands of BandBast, SOC stock would decrease by 3.05% in 2025 and up to 2.36% in 2050 under CCH1 scenarios (Table 4). As stated by Reference [35], any climate change with decreasing annual precipitation and more intensive rainfall events are likely to change soil structure and soil quality, particularly within the top soil, which may significantly affect SOC stock.

With temperature increases and decreasing rainfall in the BandBast rangelands, from 2014 onwards, SOC stock reduced significantly with the CCH1 scenarios in comparison with the no climate change scenario (P) (Figures 4 and 5). Other researchers [13,31] have indicated that increased temperatures might enhance the release of $CO_2$ to the atmosphere from SOC. Temperature increases will accelerate decomposition, and consequently increase the loss of SOC stock in the upper layer of the soil [12]. As reported by Reference [13] using the RothC model, results indicated that SOC stock will decrease by 2–6% during 40 years, under the climate change scenarios in rangeland's of southern Ireland. Using the CarboSOIL model, Muñoz Rojas et al. [10] also have shown that climate change had a negative impact on SOC contents in the upper layers of the soil section in Andalusia. They stated that land uses with little or no vegetation cover would be severely affected by climate change and provided evidence for large decreases of SOC stocks in the studied area [10]. The authors of Reference [16] also stated

that future climatic scenarios can decrease SOC stocks in the upper sections of the soil profile when rainfall decreases as compared to increases in deeper layers. Jebari et al. [8] also using the RothC model, predicted that SOC content will decrease in the different climate change scenarios in comparison with no climate change scenario for future decades overall Spain. In our study, simulations of the SOC in the rangeland of BandBast area in Ghir–O-Karzin are in agreement with the aforementioned research results since SOC decreases under climate change conditions were predicted by the model.

In addition, the RothC model predicted that more SOC will be lost due to climate change under CCH2 scenario from 2025 until 2050 (Figure 4). Under dry climate conditions, processes of soil degradation are accelerated, so that less rainfall provides lower soil moisture that results in a reduced growth and survival of vegetation and a lower storage of soil organic matter [36]. In this study, model simulation predicted that the SOC stock under the CCH1 and CCH2 scenarios showed a more declining trend than the P scenario (Figure 4). Based on these results, we could expect that the climate change may accelerates the decomposition of the SOC which is in accordance with previous studies [9,10,12,37]. Additionally, with climatic conditions from 2014 until 2050 the SOC stock will decrease by 0.24% and 5.36% under the P and CCH1 scenario, respectively (Table 4) and a significant difference ($p < 0.01$) was observed between them (Figure 5). Muñoz Rojas et al. [10] have also stated that, the absolute values cannot be directly compared among the studies due to the differences in the soil sections, therefore they compared their results with other researches based on percentage change. According to our findings however, the result of our simulations procedures (Table 4) were consistent with study of Reference [13], which projected SOC losses by 2–6% during the 40 years under the climate change scenarios in rangelands of southern Ireland.

In addition, simulations predicted the lowest SOC stock in CCH2 scenario (Figures 4 and 5) compared to the CCH1 scenario (Table 4). Furthermore, SOC stock seems to be decreasing more slowly under CCH1 than CCH2 climate change scenario (Figure 4). As a consequence, losses of SOC under CCH2 and CCH1 were 3.53% (0.57 Mg C ha$^{-1}$) and 2.36% (0.38 Mg C ha$^{-1}$) respectively (Table 4) and there was a significant difference ($p < 0.01$) between them (Figure 5).

In this study, however, the plant inputs were set at the same value in the current and future climate scenario since there are contradictory viewpoints about the effects of increase in temperature on the plant carbon inputs to soil [38,39]. Recently, Mishra et al. [40] stated that changing carbon inputs under climate change is not supported by specific scientific findings. In spite of the fact that some researchers [19] indicated that the negative effects of climate change on soil organic carbon was not significant, our study on the other hand showed that in arid rangelands BandBast climate change had a significant reduction effect on carbon storage with both scenarios (CCH1 and CCH2).

## 5. Conclusions

The RothC model estimated the SOC stock with an acceptable accuracy and therefore was used for evaluating the effect of climate change scenarios on SOC stock in the arid rangelands of Ghir–O-Karzin's BandBast. Additionally, RothC model can be used as a tool for environmental assessments related to the climatic change. The comparison of results indicates that both the two climate change scenarios (CCH1 and CCH2) significantly affected the average annual SOC stock. The study showed however, that the SOC stock trends in the CCH1 and CCH2 scenarios are more declining compared with the P scenario. The RothC simulations with the high increases in temperature and decreases of rainfall that may occur in the rangelands of BandBast predicted very high SOC stock decreases. Results of predictions obtained in this study can be used for decision-making and for the adoption of proper management practices in other arid rangelands of Iran.

**Author Contributions:** Conceptualization, B.A., S.F.A., and R.F.; Methodology, B.A., and R.F.; Software, B.A., and R.F.; Validation, B.A., and R.F.; Formal analysis, B.A., and R.F.; investigation, B.A., S.F.A., M.H.G., and R.F.; Writing—original draft preparation, B.A., and M.H.G.; Writing—review and editing, B.A., and R.F.; Project administration, B.A., S.F.A.; Funding acquisition, M.H.G.

**Funding:** This research was supported by the Shiraz University through a grant provided by the Department of Natural Resource and Environmental Engineering.

**Acknowledgments:** The authors gratefully acknowledge the Iranian University of Shiraz for the financial supports. We are highly grateful to Kevin Coleman (UK), who provided us with the RothC Model software. Additionally, authors are most grateful to M. Yazdanifar, H. Ebrahimi, A. Saeed-Mucheshi, F. Moradi, M. Romyani, Sh. Rahimi, B. Solaymani and G. Azizi for field sampling. We thank the anonymous reviewers for their comments and suggestions.

**Conflicts of Interest:** The authors declare no conflict of interest.

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
