# Peer review of "Using RothC Model to Simulate Soil Organic Carbon Stocks under Different Climate Change Scenarios for the Rangelands of the Arid Regions of Southern Iran"

_water, doi:10.3390/w11102107_

Round 1

Reviewer 1 Report

It is a very good study with overall adequate presentation of experimental results. Some additions are needed:

1) Authors should further emphasize on the novelty of their work. Why the have selected this geographical area, etc.

2) Some minor typos, grammar and syntax errors should be carefully revised and corrected accordingly.

3) Reference can be even more updated (more recent relative works).

Author Response

At the first, we have a great pleasure to review our research team's manuscript and provide valuable points. Certainly, your valuable points will enhance the richness of the manuscript.

1) Authors should further emphasize on the novelty of their work. Why the have selected this geographical area, etc.

We will make the necessary changes in the text of the manuscript. Our answers are in red color.

2) Some minor typos, grammar and syntax errors should be carefully revised and corrected accordingly.

It has been addressed.

3) Reference can be even more updated (more recent relative works).

There are many parts to be deleted. We also added some more recent references, reported in green in the text as well as at the reference list.

We hope that the provided explanations are convincing and our manuscript got the standard for printing in this journal.

Thank you for your valuable points.

With best regards.

From the authors 

Reviewer 2 Report

First of all, thank you for good research.

I would ask the author to modify the following items to help the reader understand this paper.

The organization and explanation of the paper are well established. However, a detailed explanation of the figure and the equation should add manuscript.

1) Make sure to correct the latitude and longitude clearly in Figure 1.

2) In Figure 2, modify the units of the right vertical axis to Celsius.

3) In equation 1, describe the OC(%).

Author Response

At the first, we have a great pleasure to review our research team's manuscript and provide valuable points. Certainly, your valuable points will enhance the richness of the manuscript.

Our answers are in red color.

1) Make sure to correct the latitude and longitude clearly in Figure 1. We prepared a new figure.

2) In Figure 2, modify the units of the right vertical axis to Celsius. It is changed to Celsius.

3) In equation 1, describe the OC(%). It is chanced with SOC content (%) in Eq.1.

We hope that the provided explanations are convincing and our manuscript got the standard for printing in this journal.

Thank you for your valuable points.

With best regards

From the authors

Round 2

Reviewer 1 Report

Accept as it is.

This manuscript is a resubmission of an earlier submission. The following is a list of the peer review reports and author responses from that submission.